# Research Progress on the Extraction and Purification of Anthocyanins and Their Interactions with Proteins

**DOI:** 10.3390/molecules29122815

**Published:** 2024-06-13

**Authors:** Hongkun Xue, Min Zha, Yingqi Tang, Jianduo Zhao, Xiaopeng Du, Yu Wang

**Affiliations:** College of Traditional Chinese Medicine, Hebei University, No. 342 Yuhua East Road, Lianchi District, Baoding 071002, China; xuehk0906@hbu.edu.cn (H.X.); 19567911749@163.com (M.Z.); 18631203107@163.com (Y.T.); zhao18210930562@163.com (J.Z.); 15631852526@163.com (X.D.)

**Keywords:** anthocyanins, extraction and purification, protein, interaction, application

## Abstract

Anthocyanins, as the most critical water-soluble pigments in nature, are widely present in roots, stems, leaves, flowers, fruits, and fruit peels. Many studies have indicated that anthocyanins exhibit various biological activities including antioxidant, anti-inflammatory, anti-tumor, hypoglycemic, vision protection, and anti-aging. Hence, anthocyanins are widely used in food, medicine, and cosmetics. The green and efficient extraction and purification of anthocyanins are an important prerequisite for their further development and utilization. However, the poor stability and low bioavailability of anthocyanins limit their application. Protein, one of the three essential nutrients for the human body, has good biocompatibility and biodegradability. Proteins are commonly used in food processing, but their functional properties need to be improved. Notably, anthocyanins can interact with proteins through covalent and non-covalent means during food processing, which can effectively improve the stability of anthocyanins and enhance their bioavailability. Moreover, the interactions between proteins and anthocyanins can also improve the functional characteristics and enhance the nutritional quality of proteins. Hence, this article systematically reviews the extraction and purification methods for anthocyanins. Moreover, this review also systematically summarizes the effect of the interactions between anthocyanins and proteins on the bioavailability of anthocyanins and their impact on protein properties. Furthermore, we also introduce the application of the interaction between anthocyanins and proteins. The findings can provide a theoretical reference for the application of anthocyanins and proteins in food deep processing.

## 1. Introduction

Anthocyanins, natural water-soluble pigments of flavonoids, are widely present in the vacuoles of plant flowers, fruits, stems, and leaves, giving plants different colors including red, magenta, and blue [1,2]. Compared with synthetic pigments, natural pigments have higher stability and safety. Among them, anthocyanins are one of the best natural substitutes for artificially synthesized pigments and have been used widely because of their good biological activities and safety [3]. Numerous studies have confirmed that anthocyanins exhibit various biological activities including antioxidant, anti-tumor, anti-inflammatory, hypoglycemic, vision protection, and anti-aging [4,5,6,7]. Hence, anthocyanins are widely used in fields such as food, medicine, and cosmetics. The rapid and efficient extraction of anthocyanins is an important prerequisite for their development and utilization. Therefore, developing green and efficient extraction methods is an urgent problem to be solved for the efficient utilization of anthocyanins. Currently, the extraction of anthocyanins mainly uses traditional solvent extraction methods. However, these methods exhibit some disadvantages including low extraction efficiency, long time consumption, and high solvent consumption [8]. With the continuous development of extraction technology, some advanced extraction techniques have been proposed such as ultrasound-assisted extraction, microwave-assisted extraction, enzyme-assisted extraction, ultra-high pressure-assisted extraction, and some combined-extraction techniques. However, these advanced extraction techniques also show some limits including high extraction costs and energy consumption, as well as difficulty in controlling the extraction process [9]. Thus, further in-depth research is needed on the efficient extraction methods for anthocyanins. During the extraction of anthocyanins, some impurities are also extracted, which affect the biological activities and stability of anthocyanins to a certain extent. Hence, the crude extract of anthocyanins still needs further separation and purification. Currently, the purification of anthocyanins mainly uses techniques including column chromatography, high-performance counter-current chromatography, and semi-preparative chromatography. However, these purification techniques still have certain limitations, such as low purity and preparation amounts of anthocyanins, which cannot achieve the large-scale industrial preparation of high-purity anthocyanin monomers [10]. Thus, it is crucial to develop efficient purification techniques to expand the application scope of anthocyanins. Figure 1 summarizes the different extraction and purification methods of anthocyanins from natural resources.

The chemical properties of anthocyanins are unstable and easily influenced by external factors such as temperature, pH value, oxygen, light, ascorbic acid, and metal ions, which is mainly attributed to the presence of multiple hydroxyl groups on the benzene ring of anthocyanins. Thus, the research on the stabilization of anthocyanins is currently a hot topic that urgently needs to be addressed. Previous studies have found that copigmentation, structural modification, and encapsulation strategies could significantly improve the stability of anthocyanins [1,3]. However, these methods still have problems such as low yield, poor safety, and difficulty in maintaining the color of anthocyanins. Proteins have good surface properties, self-assembly properties, and gel properties, which can form conjugates through non-covalent and covalent actions without changing the structure of anthocyanins, reducing the loss of pyran cations and thus improving the stability of anthocyanins [11]. The combination of anthocyanins and proteins can improve the functional properties of proteins including surface hydrophobicity, emulsification, foaming, etc. [11]. Thus, a better understanding of the interaction patterns between proteins and anthocyanins is crucial for their application in food systems. Currently, there are limited reports on the systematic review of extraction and purification of anthocyanins and their interaction with proteins.

Hence, the aim of this article is to (1) review systematically the extraction and purification methods of anthocyanins; (2) summarize systematically the effect of the interactions between anthocyanins and proteins on the bioavailability of anthocyanins and their impact on protein properties; and (3) introduce the application of the interaction between anthocyanins and proteins.

## 2. Structure and Properties of Anthocyanins

Anthocyanins are green, safe, and pollution-free natural flavonoid compounds widely present in flowers, fruits, stems, leaves, and roots, which give plants different bright colors. According to the Chinese National Standard for Food Additives (GB2760-2014 [12]), anthocyanins can be used as food colorants in juice, wine, jelly, and syrups [2]. In addition, anthocyanins can also be used as functional factors in health products and drugs, achieving the effect of preventing and treating various chronic diseases. Numerous studies have confirmed that anthocyanins show various biological activities including antioxidant, antibacterial, anti-tumor, hypoglycemic, vision protection, etc. [4,5,6,7]. Anthocyanins have a C6-C3-C6 carbon skeleton structure. The difference among anthocyanins is mainly due to the different types, numbers, and positions of substituents in the A and B rings and different C-cycloglycoside ligands. At present, more than 700 kinds of natural anthocyanins have been isolated from plants [13], and pelargonidin, delphinidin, petunidin, cyanidin, peonidin, and malvidin are commonly used as monomers of anthocyanins (Figure 2A). The daily per capita intake of anthocyanins in the United States and the Netherlands is about 12.5 mg and 20 mg, respectively, and the daily per capita intake of anthocyanins in males is generally higher than that in females [14]. Anthocyanins are particularly sensitive to environmental conditions such as pH, temperature, light, antioxidants, and metal ions, which can easily cause changes in the chemical structure of anthocyanins, leading to color changes and decreasing their biological activities and utilization. This is mainly attributed to the presence of a large number of phenolic hydroxyl groups and unsaturated bonds in their structure [15]. Previous reports have found that anthocyanins degrade with increasing temperature. For example, the retention rate of anthocyanins decreased from 88.19% to 28.10%, and the half-life of anthocyanins decreased from 55.0 h to 5.9 h, with an increase in temperature from 4 °C to 100 °C [15]. Figure 2B shows the structural changes in anthocyanins at different pH levels. Anthocyanins exist mainly in the form of red flavylium cations, which are easily soluble in water at a solution pH below 2. Anthocyanins exist in the form of colorless methanol off-base or hemiacetals within the pH range of 3–6 (Figure 2B). Notably, most glycosylated or glycosylated anthocyanins exhibit high color stability in the pH range of 3–6, while the color stability of anthocyanins decreases at high pH values [11]. Anthocyanins deprotonate to form purple neutral quinoid alkali with an increase in pH, and anthocyanins transform into blue ionized quinoid alkali within the pH range of 8–10 (Figure 2B). One study confirmed that anthocyanins showed high stability at pH 2–3, whereas they had poor stability under neutral and alkaline conditions (Figure 2B) [16]. Depending on the structure of anthocyanins at different pH values and their different color characteristics, anthocyanins can be developed to produce safer and more reliable pH test papers.

## 3. Anthocyanin Extraction Methods

The type of extraction technology used directly affects the extraction efficiency and biological activities of anthocyanins, which are key factors restricting the expansion of their production. Green and efficient extraction technology can prevent the degradation of anthocyanins to a certain extent, improve the extraction rate, and maintain their biological activities to a maximum extent. At present, the commonly used extraction methods for anthocyanins mainly include the solvent extraction method (SEM), ultrasound-assisted extraction (UAE), microwave-assisted extraction (MAE), ultra-high pressure assisted extraction (UHPAE), aqueous two-phase extraction (ATPE), enzymatic-assisted extraction (EAE), etc. Figure 3 exhibits the principle of different extraction and purification methods for anthocyanins.

### 3.1. SEM

SEM uses the difference in the solubility of each component in a specific solvent to achieve the purpose of extraction and separation. SEM is the most commonly used method for the extraction of anthocyanins. According to the extraction temperature, SEM can be divided into cold extraction and hot extraction. Moreover, SEM can be divided into the water extraction method and organic solvent extraction method, in which the organic solvent is generally ethanol, methanol, or acetone. In the extraction of anthocyanins, a small amount of organic acids is often added to the extraction solvent because of the instability of anthocyanins in neutral and alkaline environments. Some factors in SEM such as the material crushing degree, extraction time, extraction temperature, and solvent type can affect the extraction efficiency of anthocyanins to some extent. Currently, anthocyanins have been successfully extracted from *Lycium barbarum* and the purple potato by SEM (Table 1) [17,18]. Solvents (methanol, ethanol, and hexane) easily cause environmental pollution and do not conform to the concept of green extraction. The demand for environmentally friendly extraction solvents is also expanding with the continuous improvement in people’s awareness of decreasing environmental pollution and protecting natural resources. Alternative solvents, namely, ionic liquids (ILs), deep eutectic solvents (DESs), and natural deep eutectic solvents (NADESs), have been widely used in the extraction of natural anthocyanins. Compared with traditional extraction solvents, these alternative solvents can reduce environmental pollution more effectively and achieve a higher extraction rate. Presently, anthocyanins have been successfully extracted from mulberry and black cherry berries using DESs and NADESs, respectively (Table 1) [19,20]. In summary, SEM has some advantages including convenient operation, simple equipment, and easy implementation. However, this method has disadvantages such as time-consuming, low efficiency, and large solvent consumption. Therefore, SEM has certain limitations in the extraction of anthocyanins. Traditional solvent extraction and advanced extraction methods are usually combined to extract anthocyanins to solve the limitations of SEM.

### 3.2. UAE

UAE utilizes 20–50 kHz ultrasound to generate cavitation, thermal, and mechanical effects that damage plant cell walls, accelerate the dissolution of intracellular solutes, and thus achieve the goal of enhancing the extraction of active ingredients [21]. In addition, UAE is not limited by component polarity or molecular mass size, and UAE is suitable for the extraction of most active ingredients [22]. Many studies initially explored the effect of extraction parameters (ultrasound power, ultrasound temperature, ultrasound time, solid-to-liquid ratio, and pH) on anthocyanin yield through single-factor experiments, and then different optimization methods were used to optimize the extraction process to obtain the maximum anthocyanin yield [23]. Presently, anthocyanins have been successfully extracted from Andean blackberry, *Rubia sylvatica Nakai* fruit, blueberry (*Vaccinium ashei*) wine pomace, mulberry wine residues, and purple sweet potatoes using UAE (Table 1) [24,25,26,27,28]. In summary, UAE can improve the yield of natural anthocyanins to a certain extent. However, cavitation, thermal, and mechanical effects of ultrasound may damage the structure of anthocyanins. Therefore, ultrasound conditions (ultrasound power, extraction temperature, the solid-to-liquid ratio, and extraction time) must be strictly controlled to maintain the advantages of UAE.

### 3.3. MAE

MAE utilizes microwave radiation to directly penetrate the interior of a material. Polar molecules within a cell absorb microwave energy, causing a rapid increase in temperature. Solvent evaporation generates pressure, and the coupling effect of pressure gradient and concentration gradient promotes the transfer of target components from the inside out. In addition, the pressure generated by microwave induction can effectively rupture the plant cell wall, reduce mass transfer resistance, and enhance the yield of the target component [29]. Compared with SEM, MAE shows a higher recovery rate and efficiency. Currently, MAE technology is widely used for the extraction of anthocyanins from natural resources. Yigit et al. extracted anthocyanins from red cabbage by MAE and optimized the extraction process via RSM. Their results showed that the optimal extraction process parameters were obtained as follows: extraction time of 4 min, microwave power of 200 W, extraction temperature of 40 °C, and solid-to-liquid ratio of 1:20 g/mL. The yield of anthocyanins was 241.20 mg/L under these parameters [30]. Anthocyanins from red bean seed coats were extracted via MAE, and the extraction process was optimized using RSM. It was found that the optimal extraction conditions were obtained as follows: solid-to-liquid ratio of 1:30 g/mL, microwave power of 463 W, and microwave time of 4.2 min. The yield of anthocyanin was 32.08 mg/100 g under these conditions [31]. Moreover, anthocyanins have been successfully extracted from blackberry, cranberry, and purple sweet potatoes using MAE technology (Table 1) [32,33,34]. MAE exhibits some advantages including high extraction efficiency, low solvent consumption, simple operation, easy process control, energy conservation, and environmental protection. However, MAE generates locally high temperatures in the extraction solution, leading to the degradation of heat-sensitive components, which limits the application of MAE in the extraction field [29]. Hence, we should combine MAE with other extraction methods for the extraction of anthocyanins from natural resources to expand the application scope of microwaves in the extraction field in the future.

### 3.4. UHPAE

UHPAE uses ultra-high pressure deprotonate charged groups and destroy ionic and hydrophobic bonds, resulting in protein denaturation and conformational changes, which is conducive to easier diffusion of target components from the cell to the surrounding solvent. In addition, ultra-high pressure can increase the permeability of cell membranes and the solubility of target components, thereby improving the yield of target components. Presently, the extraction effect of anthocyanins is mainly investigated considering extraction pressure, holding time, solvent selection, the solid-to-liquid ratio, pH, and other factors. Deng et al. determined the extraction process of UHPAE anthocyanins from purple corn seeds through single-factor experiments and RSM. The results showed that the optimal extraction process was as follows: ultra-high pressure of 270 MPa, pressure time of 1.9 min, and solid-to-liquid ratio of 1:16 g/mL. The maximum extraction ratio of anthocyanins was 49.68 mg/100 g under these process conditions [35]. The optimal extraction process conditions for anthocyanins from mulberry via UHPAE were studied using single-factor experiments and RSM, and their antioxidant activity was then evaluated. The results showed that the optimal process conditions for UHPAE of mulberry anthocyanins were as follows: ethanol concentration of 75%, extraction pressure of 430 MPa, and liquid-to-solid ratio of 12:1 mL/g. The yield of anthocyanins was 1.97 ± 0.02 mg/g under these conditions. Moreover, the scavenging rate of DPPH and OH radicals increased with the increase in anthocyanin concentration. The IC_50_ values of anthocyanin extracts for scavenging DPPH and OH radicals were 0.026 mg/mL and 0.406 mg/mL, respectively [36]. Moreover, their antioxidant potential was influenced by numerous parameters intrinsic and extrinsic such as light exposure, sample pretreatment, temperature, storage, and oxygen [37]. Furthermore, UHPAE was used to extract anthocyanins from strawberry slices, blueberries, and roses (Table 1) [38,39,40]. Compared with traditional extraction techniques, UHPAE has advantages such as a shorter extraction time, higher extraction efficiency, and fewer impurities. Although UHPAE can significantly improve the yield and avoid the degradation of thermosensitive components, the extraction cost of this technology is high, which is not conducive to the large-scale industrial extraction of active ingredients.

### 3.5. ATPE

ATPE is an efficient and mild new extraction and separation technology that utilizes the difference in distribution coefficients of substances among immiscible dual aqueous phases for separation [41]. Compared with some traditional separation methods, ATPE has advantages including simpler operation, milder conditions, lower cost, and easier continuous operation. Hence, ATPE can be widely used in foods, drug analysis, and metal separation. In one study, anthocyanins from roselle were extracted by an ethanol–(NH_4_)_2_SO_4_ aqueous two-phase system, the extraction process of anthocyanins was optimized via RSM, and their antioxidant activity was evaluated [42]. It was found that the optimum conditions for the extraction of anthocyanins from roselle through ATPE were as follows: solvent system (30% ethanol and 22% ammonium sulfate), solid-to-liquid ratio of 1:30 mg/L, and pH 3.0. The yield of anthocyanins was 4.121 mg/g under these conditions. Moreover, the IC_50_ values of roselle anthocyanin extracts and the Vc values for scavenging OH radicals were 169.1 μg/mL and 448.5 μg/mL, respectively. The total antioxidant capacity of the anthocyanin extract was better than that of Vc, implying that roselle anthocyanins can be used as natural antioxidants in functional foods and medicines. Zhai et al. optimized the extraction process of ATPE anthocyanin from black soybean hull through RSM. The results showed that the optimal extraction process for anthocyanins using ATPE was as follows: solid-to-liquid ratio of 1:56 g/mL, ammonium sulfate mass fraction of 22%, ethanol of 30%, pH 3.0, and the yield of anthocyanins was 2.81 mg/g under these conditions [43]. Moreover, anthocyanins have been successfully extracted from grape skin residue, purple potatoes, and blueberries using ATPE (Table 1) [44,45,46]. All in all, ATPE has the advantages of mild conditions, high recovery, and simple operation. Thus, ATPE can be widely used as a novel extraction method for the large-scale industrial extraction of anthocyanins from natural resources.

### 3.6. EAE

EAE is a fast, green, and promising extraction method that mainly utilizes different enzymes (papain, pectinase, cellulase, etc.) to degrade plant cell walls, reduce the mass transfer resistance of target components from the inside out, increase the diffusion coefficient, and thus achieve the goal of enhancing the extraction of active ingredients [47]. In addition, EAE has advantages such as strong specificity, low energy consumption, a fast rate, high yield, and simple extract recovery [48]. Therefore, EAE is widely used for the efficient extraction of natural active ingredients. To improve the yield of anthocyanins from *Lycium ruthenicum* Murr. (LRMAs), one study completed the extraction anthocyanins by EAE and optimized the extraction process through RSM. It was found that the optimal extraction conditions for LRMAs were as follows: extraction temperature of 49 °C, ethanol concentration of 80%, extraction time of 1.05 h, and liquid-to-solid ratio of 21:1 mL/g, and the yield of LRMAs was (24.675 ± 0.027) mg/g under these extraction parameters [49]. Tan et al. optimized the ultrasound enzyme-assisted extraction of anthocyanins from grape skins using RSM combined with the genetic algorithm. The results showed that the optimum extraction process was as follows: extraction temperature of 50 °C, ultrasound power of 400 W, pectinase dosage of 0.16%, and extraction time of 0.16%, and the yield of anthocyanins was 3.01 ± 0.04 mg/g under these extraction conditions [50]. Shen et al. optimized the EAE process of anthocyanins from mulberry wine residue using Box–Behnken experiments. The results showed that the optimal extraction process was as follows: extraction time of 37 min and extraction temperature of 38 °C, and the yield of anthocyanins was 19.51 ± 0.21 mg/g under these conditions [51]. Moreover, EAE was also used to extract anthocyanins from *Acanthopanax senticosus* and *Vaccinium bracteatum Thunb*. fruits (Table 1) [52,53]. Overall, EAE has numerous extraction advantages. However, EAE has certain drawbacks including high extraction costs, easy enzyme inactivation, and difficulty in controlling the extraction conditions. Therefore, EAE is not suitable for the large-scale industrial extraction of anthocyanins. Further exploration of the EAE extraction mechanism is needed to expand its application scope in the future.

### 3.7. Other Extraction Methods

Negative pressure cavitation extraction (NPCE) is a natural product extraction method that uses cavitation, cavitation erosion, and mechanical oscillation to accelerate the rupture of plant cells and promote the continuous dissolution, release, and diffusion of active ingredients in cells. Wang et al. optimized the NPCE of anthocyanins from blueberries through single-factor experiments and a Box–Behnken design. It was found that the optimal extraction conditions were as follows: solid-to-liquid ratio of 1:30 g/mL, ultrasound power of 0.36 W/cm^2^, ethanol concentration (*v*/*v*) of 68.61%, negative pressure of −0.07 Pa, and extraction time of 15 min. The contents of total phenols, flavonoids, and total anthocyanins were 22.5 μg/mL, 30.0 μg/mL, and 32.5 μg/mL, respectively, under the optimal extraction conditions [54]. Compared with traditional extraction technology, NPCE has some advantages including mild reaction conditions, a high extraction rate, low energy consumption, simple equipment, and large-scale industrial production. Pulsed electric field extraction (PEF) is a natural product extraction method that uses PEF to generate cell electroporation and improve the extraction ratio of target substances. Currently, HVPEF has been used for the extraction of proteins, pigments, sugars, and other active substances [55,56]. Puertolas et al. extracted anthocyanins from purple-fleshed potatoes by PEF and optimized them through RSM. It was found that the optimal extraction process for anthocyanins was as follows: extraction temperature of 40 °C, electric field intensity of 3.4 kV/cm, treatment time of 31 min, and ethanol concentration of 96%, and the yield of anthocyanins was 63.9 mg/100 g under these conditions [57]. Compared with the traditional extraction technology, PEF exhibits some merits including a lower extraction temperature, faster extraction efficiency, higher yield, and quality. Supercritical CO_2_ extraction is a method to extract natural products using CO_2_ between liquid and gaseous states as extraction solvent under certain conditions. The fluid above the critical temperature and pressure is called supercritical fluid, which has strong fluidity, transfer, and dissolution ability. The solubility of anthocyanins can be improved by adding an entrainment agent [58]. The supercritical CO_2_ extraction method has a low extraction temperature and high efficiency. Therefore, this method is suitable for extracting thermosensitive components such as anthocyanins and chlorophyll. Paes et al. extracted anthocyanins from blueberry residues by using supercritical CO_2_ extraction technology and optimized the extraction process via RSM. The results showed that the optimal extraction process for anthocyanins was as follows: extraction pressure of 10 MPa, extraction temperature of 40 °C, and solid-to-liquid ratio of 1:25 g/mL, and the yield of anthocyanins was 22.2 mg/g under the optional extraction conditions [59]. Compared with the solvent extraction method, the supercritical CO_2_ extraction method has some advantages including higher safety, higher extraction purity, and less extraction material loss. The microbial fermentation extraction method uses enzymatic catalysis related to microbial action to destroy plant cell wall damage, promote the release of anthocyanins into the extraction solution, and accelerate the efficiency and rate of extraction. The microbial fermentation extraction method uses enzymatic catalysis related to microbial action to destroy the cell wall of the extracted raw material cells and promote the release of anthocyanins into the extraction liquid to accelerate the extraction efficiency and rate. Moreover, microbial fermentation can decompose macromolecular impurities such as polysaccharides and proteins into small molecules, which greatly reduces the difficulty of extract purification [60]. Zhang et al. extracted anthocyanins from concentrated purple sweet potato juice by microbial fermentation. The results showed that fermented purple potato juice had high DPPH free radical scavenging ability and stability [61] (Table 1). The fermentation method addresses the shortcomings of traditional extraction methods such as a low extraction rate, difficult purification, and a low utilization rate of raw materials. In addition, microorganisms can also act on the residue after pigment extraction and then produce by-products (alcohol), which greatly reduces the cost of industrial production by improving the utilization rate of raw materials. A comprehensive analysis of all extraction methods found that the single-extraction method has the disadvantages of low extraction efficiency and high extraction costs. In the actual production process, people always expect to achieve the best yield and production efficiency. Therefore, we should try to combine and work with different extraction methods to give full play to their respective advantages and thus achieve the best extraction and production effect of anthocyanins and solve the bottleneck problem of expanding the application of anthocyanins. Currently, anthocyanins have been successfully extracted from purple potatoes, *Rhodomyrtus tomentosa*, and blueberry pomace by the ultrasonic-microwave combined method, the cellulase-microwave combined method, and deep eutectic solvent combined with ultrasound technology, respectively [62,63,64] (Table 1). Combined extraction technology can improve the extraction efficiency of anthocyanins to a certain extent, which lays a foundation for further development and utilization of anthocyanins. However, combined extraction technology still has some limitations, such as a complex extraction process, imprecise control, and an unclear extraction mechanism. Thus, it is necessary to further explore the combined extraction process and extraction mechanism in the future.

**Table 1 molecules-29-02815-t001:** Extraction solvents, methods, and process conditions for the extraction of anthocyanins using different extraction methods.

Source	Extraction Solvent	Extraction Method	Extraction Process	Yield	References
*Lycium ruthenicum* Murr	60% ethanol	SEM	Extraction temperature 48 °C, extraction time 4 min, static extraction pressure 8 MPa, cycle two times	19.89 mg/g	[17]
Purple potato (*Ipomoea batatas* L.)	77% ethanol	UHPAE	Extraction temperature 67 °C, extraction time 25 min, solid-to-liquid ratio 1:58 g/mL	200.13 mg/100 g	[18]
Mulberry (*Morus alba* L.)	ChCl-HL ratio 2:57 (*v*/*v*)	DESs	Extraction temperature 24.31 °C, extraction time 54.10 min	76.60 mL/g	[19]
*Aronia melanocarpa*	Choline chloride-glycerol	UMAE	Extraction temperature 58–64 °C, extraction time 60–105 min, solid-to-liquid ratio 1:30–1:40 g/mL	448.87 mg/g	[20]
Andean blackberry (*Rubus glaucusBenth.*)	80% ethanol and 0.1% acidified hydrochloric acid	UAE	Extraction temperature 40 °C, ultrasound power 110 W, extraction time 15 min, solid-to-solvent ratio 1:10 g/mL	162.00 mg/L	[24]
*Rubia sylvatica* Nakai	30% ethanol	UAE	Extraction temperature 55 °C, pH 3.0, ultrasound power 400 W, liquid-to-solid ratio 20 mL/g, extraction time 20 min	22.35 ± 0.89 mg /g	[25]
Blueberry wine pomace (*Vaccinium ashei*)	70% ethanol, and 0.01% hydrochloric acid	UAE	Extraction temperature 61.03 °C, liquid-to-solid ratio 21.70 mL/g, extraction time 23.67 min, ultrasound power 400 w	4.27 mg/g	[26]
Mulberry wine residues (*Morus alba* L.)	80% ethanol	UAE	Extraction temperature 52 °C, ultrasound power 315 W, enzyme dosage 0.22%, extraction time 94 min	5.98 mg/g	[27]
Purple sweet potato (*Ipomea batatas* L.)	83% polyethylene glycol (200)	UAE	Extraction temperature 64 °C, extraction time 80 min	83.78 mg/100 g	[28]
Red cabbage (*Brassica oleracea* L. var. *capitata.* f. *rubra*)	50% ethanol	MAE	Extraction time 5 min, solid-to-liquid ratio 1:20 g/mL, microwave power 200 W	241.20 mg/g	[30]
Adzuki bean (*Vigna angularis*) seed coat	30% ethanol	MAE	Microwave power 463 W, solid-to-liquid ratio 1:30 g/mL, extraction time 42 min	32.08 mg/100 g	[31]
Blackberry (*Rubus fruticosus*)	52% ethanol	MAE	Microwave power 469 W, liquid-to-solid ratio 25:1 g/mL, microwave time 4 min	2.18 ± 0.06 mg/g	[32]
Cranberry (*Vaccinium macrocarpon Ait.*)	52% ethanol	MAE	Extraction temperature 50 °C, extraction time 8 s, solid-to-liquid ratio 1:28 g/mL	3.06 ± 0.05 mg/g	[33]
Purple sweet potato (*Ipomea batatas* L.)	30% ethanol	MAE	Solid-to-liquid ratio 1:3 g/mL, microwave power 320 W, extraction time 500 s	31.16 mg/100 g	[34]
Purple corn (*Zea mays* L.) seeds	Anhydrous ethanol	UHPAE	Extraction pressure 270 MPa, ultra-high pressure time 1.9 min, ultrasound time 8.5 min	49.68 mg/100 g	[35]
Mulberry (*Morus alba* L.)	75% ethanol	UHPAE	Extraction pressure 430 MPa, liquid-to-solid ratio 12:1 mL/g	1.97 ± 0.02 mg/g	[36]
Vacuum freeze-dried strawberry (*Fragaria x ananassa Duch.*) slices	Anhydrous ethanol	UHPAE	Extraction pressure 100 MPa, ultra-high pressure time 5 min, ultrasound time 25 min	304.39 mg/kg	[38]
Blueberry (*Vaccinium corymbosum* L.)	57% ethanol	UHPAE	Extraction pressure 187 MPa, extraction time 6 min, solid-to-liquid ratio 1:29 g/mL	5.16 ± 0.12 mg/g	[39]
Rose (*Rosa hybrida*)	13.2% citric acid	UHPAE	Extraction pressure 400 MPa, extraction time 6 min, solid-to-liquid ratio 1:25 g/mL	1089.42 mg/100 g	[40]
Roselle (*Hibiscus sabdariffa* L.)	40% ethanol and 0.24 g/mL(NH_4_)_2_SO_4_	ATPE	Liquid-to-solid ratio 40:1 mL/g, extraction temperature 40.5 °C, extraction time 24.5 min	4.12 mg/g	[42]
Black soybean hull (*Glycinemax* (L.) Merr.)	30% ethanol and 22% (NH_4_)_2_SO_4_	ATPE	Solid-to-liquid ratio 1:56 g/mL, pH 3.0	2.81 mg/g	[43]
Aqueous grape (*Vitis vinifera* L.) pomace	40% ethanol and 26% (NH_4_)_2_SO_4_	ATPE	Solid-to-liquid ratio 1:38 g/mL, pH 3.0	3.05 ± 0.07 mg/g	[44]
Purple sweet potato (*Ipomea batatas* L.)	25% ethanol and 22% (NH_4_)_2_SO_4_	ATPE	Liquid-to-solid ratio 45:1 mL/g, pH 3.3	311.00 mg/100 g	[45]
Blueberry (*Vaccinium corymbosum* L.)	24% ethanol and 18% (NH_4_)_2_SO_4_	ATPE	Ultrasound power 300 W, solid-to-liquid ratio 1:30 g/mL, extraction time 60 min	2.14 ± 0.05 mg/g	[46]
*Lycium ruthenicum* Murr	80% ethanol	EAE	Extraction temperature 49 °C, extraction time 1 h, liquid-to-solid ratio 21:1 mL/g	24.68 ± 0.03 mg/g	[49]
Grape (*Vitis vinifera* L.) skins	60% ethanol	EAE	Extraction temperature 50 °C, ultrasound power 400 W, pectinase dosage 0.16%, extraction time 28 min	3.01 ± 0.04 mg/g	[50]
*Lycium ruthenicum* Murr	Anhydrous ethanol	EAE	Extraction temperature 38 °C, extraction time 37 min	19.51 ± 0.21 mg/g	[51]
*Acanthopanax senticosus* dried fruit	Methanol	EAE	Liquid-to-solid ratio 18:1 mL/g, pectinase 4.2%, digestion temperature 55 °C, digestion time 3.0 h	6.00 mg/g	[52]
*Vaccinium bracteatum* Thunb. Fruit	50% ethanol	EAE	Cellulase–pectinase ratio 2:1, pH 4.0, solid-to-liquid ratio 1:30 g/mL, digestion temperature 50 °C, digestion time 180 min	136.08 mg/100 g	[53]
Purple-fleshed potato (*Ipomea batatas* L.)	96% ethanol	PEFAE	Electric field strength 3.4 kV/cm, processing time 105 s, extraction temperature 40 °C, extraction time 480 min	65.80 mg/100 g	[57]
Blueberry (*Vaccinium corymbosum* L.)	5% ethanol	Subcritical liquid and supercritical CO_2_ extraction	Extraction temperature 40 °C, extraction pressure 20 MPa, solvent flow rate 10 mL/min	19.60 mg/g	[59]
Purple sweet potato (*Ipomea batatas* L.)	Anhydrous ethanol	Microbial fermentation	Initial sugar content 150 g/L, fermentation time 4 d, extraction temperature 29.8 °C, pH 4.10, amount culture added 0.16%	8.80 mg/g	[61]
Purple potato (*Ipomea batatas* L.)	65% ethanol	UMAE	Ultrasonication time 24 min, liquid-to-solid ratio 21:1 mL/g, ultrasound power 210 W, microwave time 1 min, microwave power 500 W	1.11 mg/g	[62]
*Rhodomyrtus* *Tomentosa*	50% ethanol	Cellulase–microwave combined method	Cellulase addition 2.8%, liquid-to-solid ratio 21:1 mL/g, microwave power 494 W, microwave time 6.5 min	56.98 mg/100 g	[63]
Blueberry (*Vaccinium corymbosum* L.) pomace	Choline chloride– 1, 4-butanediol (molar ratio of 1:3)	Deep eutectic solvent combined with ultrasound technology	Moisture content 29%, extraction temperature 63 °C, liquid-to-solid ratio 36:1 mL/g	11.40 ± 0.14 mg/g	[64]

Abbreviations: solvent extraction method (SEM); ultrasound-assisted extraction (UAE); microwave-assisted extraction (MAE); ultra-high pressure-assisted extraction (UHPAE); aqueous two-phase extraction (ATPE); enzymatic-assisted extraction (EAE); deep eutectic solvents (DESs); ultrasonic–microwave-assisted extraction (UMAE); pulsed electric field-assisted extraction (PEFAE).

## 4. Purification of Anthocyanins

During the extraction process, anthocyanins and a large number of impurities (soluble sugars, proteins, minerals, and cellulose) are extracted simultaneously. Excessive impurities will affect the biological activities, stability, and final product quality of anthocyanins. Hence, the separation and purification of anthocyanin crude extracts is an essential step to obtain anthocyanins with high stability and strong biological activities. At present, the main purification methods for anthocyanins include column chromatography, membrane separation (MS), high-speed counter-current chromatography (HSCCC), and preparative high-performance liquid chromatography (PHPLC).

### 4.1. Column Chromatography

Currently, column chromatography is one of the most commonly used methods for purifying anthocyanins. Column chromatography can be divided into gel column chromatography, silica gel chromatography, ion exchange resin chromatography, polyamide chromatography, and macroporous resin chromatography based on different stationary phases. The macroporous adsorption resin method is a purification method that was developed in recent years. Its basic principle is to use macroporous adsorption resin to adsorb trace amounts of hydrophilic phenolic derivatives from dilute aqueous solutions and then wash and recover them to remove impurities, thereby achieving the purpose of purification. The adsorption capacity of different kinds of macroporous adsorption resins is also different. Yang et al. enriched anthocyanins from roselle by macroporous resin and analyzed the dynamic adsorption of the resin via a mass transfer zone (MTZ) model. The results indicated that the main anthocyanins included delphinidin-3-O-sambubioside and cyanidin-3-O-sambubioside. After purification with macroporous resin, the content and purity of anthocyanins increased by 6 times and 4 times, respectively [65]. Xu et al. purified purple potato anthocyanins by AB-8 macroporous resin and optimized the purification process. The results showed that the optimal purification parameters of anthocyanins in purple potatoes were as follows: sampling concentration of 0.5 mg/mL, pH of 2.0, sampling flow rate of 1 mL/min, elution concentration of 70%, pH of 1.0, and elution flow rate of 1 mL/min. Under these conditions, the specific absorbance of anthocyanins after purification was 8.43 times higher than that before purification [66]. Guo et al. used macroporous resin X-5 to purify anthocyanins from black currant residue and optimized the purification conditions through orthogonal experiments. It was found that the optimal purification process was as follows: pH of 2.0, adsorption time of 6 h, ethanol solution of 70%, and desorption time of 6 h. The purity of anthocyanins after purification was 20.9 times higher than that before purification under these conditions [67]. Tian et al. and Xue et al. purified anthocyanins from cherries and blueberries with Sephadex LH-20. The results showed that the purity of purified anthocyanins by this technique was 96% and 90.96%, respectively [68,69]. Wang et al. obtained the purification process parameters of anthocyanins from blackcurrant residue through X-5 macroporous resin as follows: diameter-to-height ratio of 1:20, mass concentration of adsorption solution of 34 mg/mL, pH of 2.0, adsorption volume of 25 mL, adsorption flow rate of 1 mL/min, elution volume of 4 BV, ethanol volume fraction of 60%, and desorption flow rate of 1 mL/min. The recovery rate of anthocyanins was 89.80% under these conditions. Compared with before purification, the color value of the purified anthocyanins increased by 18.26 times [70]. The purification effect of macroporous resin on anthocyanins is related to the type, specific surface area, and polarity of macroporous resin. The loading amount, eluent concentration, pH, and eluent flow rate affect the adsorption amount, desorption rate, and purity of anthocyanins. Based on the different characteristics of the raw materials and the feasibility of experimental conditions, suitable resins are selected to improve the purity of anthocyanins. However, column chromatography has difficulties in achieving the large-scale separation and purification of anthocyanins because of its small preparation quantity and low purity.

### 4.2. Membrane Separation

Membrane separation is the separation of the target component using a natural or artificially prepared selective permeable membrane. It applies a driving force (such as pressure difference, concentration difference, potential difference, etc.) on both sides of the membrane to concentrate and purify the solvent and solute, thus achieving the separation and purification of the target component. Currently, the membranes used for purification of anthocyanins mainly include micro-filtration membranes (MF), ultra-filtration membranes (UF), and nano-filtration membranes (NF). The whole process of membrane separation technology is physical and does not involve chemical reactions. In addition, membrane separation technology has the advantages of a wide application range, health and environmental protection, acid and alkali resistance, and low energy consumption. Currently, membrane separation technology has been widely used in different fields including seawater desalination, food, medicine, and health. He et al. purified anthocyanins from black rice crude extracts through MS and resin adsorption purification technology. The optimal conditions for purification of anthocyanins were as follows: liquid-to-solid ratio of 12:1 mL/g, temperature of 50 °C, time of 80 min, pH of 3.2, and the yield of anthocyanin reached 0.99 ± 0.01% under these conditions. Moreover, after elution with 85% ethanol solution and purification with a 200 Da molecular weight membrane, the recovery and purity of anthocyanins were 0.86% and 95.93%, respectively [71]. He et al. isolated and identified the anthocyanins from blackcurrant polyphenols (BPs) using an ultrafiltration membrane with a molecular mass of 1 kDa. The results showed that BPs with less than 1 kDa mainly contained six anthocyanins. In addition, anthocyanins in BPs showed better biological activities than crude extracts [72]. Chandrasekhar et al. purified anthocyanins from blackberry fruits by membrane separation. The concentration of anthocyanins increased from 397.78 mg/L to 768.89 mg/L before purification [73]. Cisse et al. used UF and NF to concentrate anthocyanins in roselle extract, and the concentration increased from 4 g/100 mg to 25 g/100 mg [74]. Munoz et al. evaluated the retention rate of anthocyanins in grape marc extract by three different NF membranes. The results showed that the retention rate of anthocyanins was higher than 99.42% under three different NF membranes [75]. In summary, membrane separation can significantly improve the purity of anthocyanins. Moreover, membrane separation is an energy-saving and green purification method. However, membrane separation also shows some disadvantages including low purification efficiency and purification and high costs. Thus, membrane separation has not been applied to the separation and purification of natural anthocyanins on a large scale.

### 4.3. HSCCC

HSCCC is a liquid–liquid chromatography partitioning technique. HSCCC uses the difference in the distribution coefficient between the two phases to achieve separation. HSCCC can avoid the irreversible adsorption of trace active ingredients on the solid phase carrier [76]. Moreover, the impurities in the target fraction after HSCCC separation and purification are significantly reduced, and the probability of successfully separating active compounds is greatly improved. Currently, HSCCC is widely used to prepare bioactive compounds from natural plant resources. Methyl tert-butyl ether–n-butanol–acetonitrile–water–trifluoroacetic acid (1:4:1:5:01, *v*/*v*) was used as a biphasic aqueous system to purify anthocyanins from bilberry by HSCCC. Two anthocyanin monomers including delphinidin-3-O-sambubioside and cyanidin-3-O-sambubioside were obtained through HSCCC, and their retention rates were 26.0% and 15.4%, respectively [77]. Li et al. purified anthocyanins from red grape by HSCCC. The results showed that the purity values of three anthocyanins (delphinidin-3-O-glucoside, malvidin-3-O-glucoside, and peonidin-3-O-glucoside) obtained via HSCCC were 93.7%, 95.2%, and 91.6%, respectively [78]. Xue et al. purified mulberry anthocyanins by HSCCC. The results showed that the purity values of three anthocyanins (delphinin-3-glucoside, cyanin-3-glucoside, and pelargonin-3-glucoside) obtained by HSCCC were 92.27%, 94.05%, and 90.82%, respectively [79]. In addition, HSCCC was used to purify anthocyanins from *Acanthopanax Sessilicum* and blueberries. The results showed that cyanidin-3-xylosyl-galactoside with a purity of 95.64% and cyanin-3-glucoside with a purity of 94.16% could be obtained by HSCCC, respectively [80,81]. Based on the above analysis, HSCCC shows some merits including simplicity, high preparation capacity and separation efficiency, good reproducibility, no irreversible adsorption, low cost, and complete sample recovery. Thus, HSCCC is widely used for the separation and purification of anthocyanins from natural resources.

### 4.4. PHPLC

The principle of PHPLC is to utilize differences in the physicochemical properties of each component in a mixture, which allows them to be distributed to varying degrees in two immiscible phases. Each component can undergo multiple distributions during the relative motion of the two phases, thereby achieving the purpose of separation [82]. PHPLC separation and purification of crude extracts must consider the purity of the target material, yield, operating composition, and production cycle. PHPLC is a common separation and analysis method that has been widely used in research fields such as food analysis and the purification of active ingredients. Malvidin-3,5-O-diglucoside and malvidin-3,5-O-diglucoside coumaroyl were obtained by semi-PHPLC, and their purity was 99.54% and 98.28%, respectively [83]. Liu et al. prepared delphinin-3-O-galactoside and malvidin-3-O-galactoside from blueberries through semi-PHPLC, and their purity values were 96.98% and 95.63%, respectively [84]. Wang et al. isolated and purified anthocyanins from blueberries by semi-PHPLC and obtained four anthocyanins monomers (delphinidin, cyanidin, petunidin, and malvidin) with purity values of 98.2%, 96.3%, 92.6%, and 90.5%, respectively [85]. Sweet cherry anthocyanins were purified by PHPLC to obtain two anthocyanins monomers including cyanidin-3-rutoside, with a purity of 97%, and cyanidin-3-glucoside, with a purity of 98% [86]. Moreover, Chorfa et al. prepared five common anthocyanin monomers (cyanidin, delphinidin, pelargonin, malvidin, and petunidin) from wild blueberries using PHPLC, and the purity of these monomers was above 99% [87]. All in all, PHPLC has the advantages of good separation and purification effect, high detection accuracy, wide application range, and automatic continuous separation [88]. Thus, PHPLC is widely used in the separation and purification of high-value-added products such as organic acids, bioactive bases, and flavonoids.

### 4.5. Combined Purification Method

The types and structures of anthocyanins are complex. The single purification method is not effective for the purification of anthocyanins from natural resources. Currently, many studies purify anthocyanins and improve their purity using combined purification methods. Xue et al. separated and purified anthocyanins from blackcurrants using AB-8 macroporous resin–Sephadex LH-20 gel column chromatography to obtain two anthocyanin monomers (delphinin-3-glucoside and cyanidin-3-rutinoside) [89]. Yu et al. isolated and purified anthocyanins from blueberries using macroporous resin–medium pressure column chromatography to obtain cyanidin-3-O-glucoside with a purity of 90.88% [90]. Chen et al. separated and purified anthocyanins from mulberries by solid phase extraction combined with high-performance liquid chromatography. They finally obtained three anthocyanins, namely, cyanidin-3-glucoside, cyanidin-3-rutinoside, and pelargonin-3-glucoside, and their retention rates were 67.52%, 31.29%, and 1.06%, respectively [91]. Different purification methods have their advantages (effective, automatic continuous separation, and environmentally friendly) and disadvantages (high equipment requirements and process optimization standards). Hence, we should actively explore new and efficient purification methods for improving anthocyanin purification.

## 5. Effects of Proteins on the Stability of Anthocyanins

Previous reports have confirmed that anthocyanin-rich foods are easily affected by harsh environments such as high temperature, pressure, and pH during processing, resulting in decreased anthocyanin stability [92]. Non-covalent and covalent interactions between anthocyanins and proteins can prevent the degradation of anthocyanins. Proteins have a certain binding ability to anthocyanins. However, different proteins have certain differences in affinity for different anthocyanins [93,94,95].

### 5.1. Protein Interacts with Anthocyanins

Proteins and anthocyanins can be combined through covalent enzymatic and non-enzymatic interactions (alkaline reactions and free radical grafting) or non-covalent interactions (hydrogen bonding, hydrophobic interactions, electrostatic forces, van der Waals force forms) to protect the stability of natural anthocyanins during processing and storage [96,97]. Figure 4 shows the non-covalent interactions between anthocyanins and proteins. Table 2 displays the interaction forces, effects, and binding energy between proteins and anthocyanins.

#### 5.1.1. Natural Proteins

Proteins and anthocyanins often coexist in food processing systems. The binding between proteins and anthocyanins is mainly formed by weak forces such as hydrogen bonds and hydrophobic interactions between amino acid side chains and anthocyanins. Multispectral and molecular docking methods can be used to determine the interaction between the two forces. Chamizo-Gonzalez et al. evaluated the interaction between grape seed 7S globulin and grape malvidin-3-glucoside using molecular docking and dynamics simulation. The results showed that 7S globulin interacted with anthocyanins through hydrogen, alkyl, and π-alkyl bonds [98]. β-casein can be used as the most potential protein carrier of cyanidin-3-O-glucoside (C3G), and this is attributed to the large number of hydrogen bonds and interaction energy between β-casein and anthocyanins, which can effectively improve the stability and bioavailability of anthocyanins [99]. The underlying interaction mechanisms of purple potato anthocyanins with casein (CA) and whey protein (WP) were investigated using multispectral technology and molecular docking simulation methods. It was found that CA and WP non-covalently bound to anthocyanins from purple potatoes through different binding sites, and their main forces were hydrogen bonding and van der Waals forces [100]. Whey protein isolate (WPI) interacted with malvidin-3-O-galactoside (M3G) through hydrogen bonding and hydrophobic interaction, thereby improving the stability and antioxidant activity of anthocyanins [101]. M3G bound to bovine serum protein through hydrogen bonding and electrostatic interactions to improve anthocyanins stability [102]. Meng et al. analyzed the interaction between C3G and β-lactoglobulin (β-LG) by fluorescence quenching spectroscopy. The results showed that C3G spontaneously combined with β-LG mainly through hydrophobic interactions, thus improving the bioavailability and stability of C3G [103]. In summary, proteins can form a spatial accumulation effect to inhibit the destruction of anthocyanin pyranic cations by exogenous conditions. Compared with specific embeddings, this ordinary embedding has both wrapping and embedding effects in space. However, the large spatial gaps between molecules in this effect make it difficult to form complete encapsulation. Anthocyanin molecules are easily detached from the embedding system under these conditions. Thus, it is necessary to screen out the component molecules with a high matching binding degree through quantum chemistry computational simulation and design a targeted protection system [104]. This spatial protection has strong spatial matching and specificity, which is conducive to accurately fixing anthocyanins in the molecular protection domain (protective chamber) of proteins and then extracting their stability and biological activities. The spatial matching degree between the pocket domain and anthocyanins can be further calculated through molecular dynamics in theory, which contributes to quickly screening out the possible action pockets of anthocyanins and then expanding the application range of anthocyanins and macromolecules.

#### 5.1.2. Modified Proteins

Compared with natural proteins, modified proteins have better effects on the stability and biological activities of anthocyanins. Previous reports found that WP products generated by the Maillard reaction could effectively improve the thermal stability of C3G under different pH conditions [105]. Chen et al. investigated the interaction between preheated soybean protein isolate (PSPI) and C3G. The results showed that PSPI had a strong binding affinity with C3G, which was attributed to the fact that the preheating treatment could effectively change the secondary structure of PSPI, resulting in the easy binding of PSPI with C3G [106]. Preheating WP (50 °C) and CA (60 °C) was found to improve the thermal stability, oxidation, and photostability of anthocyanins from grape skin more effectively than natural protein [107]. Attaribo et al. studied the interaction between preheated silkworm pupa protein (P-SPP) and C3G and its effect on the stability of C3G. The results showed that P-SPP at 80 °C could effectively reduce the degradation rate and increase the thermal stability of C3G [108]. Zang et al. investigated the interaction mechanism between preheated whey protein isolate (P-WPI) and anthocyanins using multispectral and molecular dynamics. Compared with natural whey protein isolate, P-WPI could more effectively protect the stability of anthocyanins, which was attributed to the fact that the interaction between P-WPI and anthocyanins could change the secondary structure of P-WPI [109]. Ren et al. studied the influence of the preheating temperature of whey protein (WP) on the color expression and stability of anthocyanins in the presence of ascorbic acid. The results showed that P-WP and N-WP had different effects on the half-life of anthocyanins. Compared with N-WP (10 mg/mL), P-WP had no more significant effect on the half-life of anthocyanins of purple corn, grape, or black carrot. This was due to the high folding degree of N-WP at 40 °C or 50 °C, which was more conducive to the interaction between anthocyanins and ascorbic acid, thereby reducing anthocyanin color loss [110].

### 5.2. Effects of Proteins on the Bioavailability of Anthocyanins

The low bioavailability of anthocyanins is a major problem limiting their application [111]. Anthocyanins start in the mouth (pH 5.6~7.9), and a portion of anthocyanins are converted to the chalcone form in neutral/weakly alkaline conditions. In addition, microbes in the mouth can also break anthocyanins into small molecular compounds. Protocatechulic acid and phloroglucinol were proven to be the main degradation products of anthocyanins [112]. The oral degradation of anthocyanins can be negligible because of their short digestion time. Anthocyanins mainly exist in flavylium cationic forms in the stomach environment (pH 1.5~3.5) [111]. A large contact area facilitates the entry of anthocyanins into the hepatic intestinal circulation in the intestine (pH 6.7~7.4), and the digestive enzymes and microbiota in the intestine can decompose and metabolize anthocyanins into phenolic metabolites [113,114]. After metabolism by the liver, anthocyanins may return to the intestinal system through bile or enter circulation, where they are subsequently cleared by the kidneys and excreted in the urine [115]. Preventing the conversion from a colored structure to a colorless structure is a key factor in maintaining the stability of anthocyanins (Figure 5) [116,117]. Proteins can protect anthocyanins from enzymatic and oxidative degradation during ingestion and assimilation [118]. Anthocyanin-protein complexes can protect anthocyanins from degradation in the stomach, control the release of anthocyanins in the intestine, and increase the content of anthocyanins in the intestine and colon. *In vitro*-simulated digestion is a method that is currently used to determine anthocyanin bioaccessibility and explore the potential for anthocyanins to interact with (and be absorbed by) proteins [117]. Moreover, a previous report found that anthocyanins readily degraded into small phenolic substances (protocatechuic acid and ferulic acid) in a simulated intestinal environment [116]. Protein was found to inhibit the loss of anthocyanin bioavailability [118]. CA, WPI, and bovine serum albumin (BSA) were found to protect anthocyanin stability during simulated digestion *in vitro* and improve the biological activities of anthocyanins [118,119]. Moreover, WP was found to protect anthocyanins from gastric digestion, promote their release into the intestine, and improve the bioavailability of anthocyanins [100]. Lang et al. investigated the effects of adding α-casein and β-casein to a simulated digestive system on the stability, antioxidant capacity, and bioavailability of blueberry anthocyanins. The results showed that both α-casein and β-casein could improve the stability of blueberry anthocyanins during intestinal digestion and protect their antioxidant capacity. In addition, the addition of α-casein or β-casein could deliver more anthocyanins to the colon, improving bioavailability [118]. Anthocyanins are easily separated from protein–anthocyanin particles during digestion for subsequent intestinal absorption and enhanced bioavailability [120]. In summary, anthocyanins wrapped in protein protective cavity can effectively inhibit the degradation of anthocyanins in the digestive tract environment and improve anthocyanin bioavailability.

**Table 2 molecules-29-02815-t002:** The interaction forces, effects, and binding energy between proteins and anthocyanins (“↓”, decrease; “↑”, increase).

Anthocyanin	Proteins	Interaction Forces	Interaction Effect	Binding Energy	References
Mv3glc	*Vitis vinifera* grape seed 7S globulin	Hydrogen, alkyl, and π-alkyl	The color stability of anthocyanins ↑	________	[98]
C3G	CM proteins (α-LA, β-LG, α_s1_-CA, and β-CA)	Hydrogen bonding	The stability and bioavailability of anthocyanins ↑	________	[99]
Purple potato (*Ipomea batatas* L.) flour	CA and WP	Hydrogen bonding and van der Waals forces	The contents of α-helix and β-turn ↓; The contents of β-sheet and irregular coil ↑	CA-Pt3G (−10.29 kJ·mol^−1^); CA-Pn3G (−10.44 kJ·mol^−1^), WP-Pt3G (−10.23 kJ·mol^−1^), and WP-Pn3G (−9.21 kJ·mol^−1^)	[100]
M3G	WPI	Hydrogen bonding and hydrophobic forces	The contents of α-helix ↓; the contents of β-sheets ↑	−141.30 kcal/mol	[101]
M3G	BSA	Electrostatic interactions and hydrogen bonding	The contents of α-helix ↑; the contents of β-sheet, turn, and random coil ↓	________	[102]
C3G	β-LG	Hydrophobic interactions	The antioxidant capacities of C3G ↑	________	[103]
C3G	SPI	Hydrophobic interactions	The contents of α-helix and no regular curl ↓; the contents of β-folding and cornering ↑	________	[106]
Grape (*Vitis vinifera* L.) skin	WP	Hydrogen bonding and hydrophobic interactions	The thermal, oxidation and, photo stability of anthocyanins ↑	________	[107]
C3G	SPP	Hydrophobic interactions	The contents of α-helix ↓; the contents of β-sheet and β-turn ↑	________	[108]
M3G	WPI	Hydrogen bonding	The stability of anthocyanins ↑	________	[109]
C3G	OVA	Hydrogen bonding and van der Waals forces	The contents of α-helix ↑; the contents of β-turn and random coil ↓	________	[121]
Black rice (*Oryza sativa* L.)	Rice (*Oryza sativa* L.) protein	Hydrophobic and hydrogen bonding	The contents of β-sheet ↑	________	[122]
Black rice (*Oryza sativa* L.)	WPI	Hydrophobic interactions	The emulsifying properties of the WPI-AN ↑	________	[123]
Black rice (*Oryza sativa* L.)	SPI	Hydrogen bonding	The contents of α-helix ↑; the contents of β-sheet ↓	________	[124]
C3G	SPI	Hydrophobic interactions	The contents of α-helix and random coil ↓; the contents of β-sheet ↑	________	[125]
Rose (*Rosa hybrida*)	WPI	Hydrogen bonding and van der Waals forces		________	[126]
C3G	SP	Hydrogen bonding and hydrophobic interactions	The surface hydrophobicity and thermostability of soy protein ↓	−6.8072 Kcal/mol	[127]
C3G	β-Lg and β-CA	Hydrophobic interactions	The contents of α-helix and β-sheet ↓	________	[128]
C3G	α-CA	Hydrogen bonding and van der Waals forces	α-helix ↑; β-folding and cornering ↓; no regular curl ↑	________	[129]
	β-CA	Electrostatic gravity	α-helix ↑; β-folding and cornering ↑; no regular curl ↓		
	WP	________	No significant changes		
	β-LG		α-helix ↓; β-folding and cornering ↑		

Abbreviations: α-LA, α-lactalbumin; α_s1_-CA, α_s1_-casein; α-CA, α-casein; β-CA, β-casein; β-LG, β-lactoglobulin; Mv3glc, malvidin-3-glucoside; M3G, malvidin-3-O-galactoside; C3G, cyanidin-3-O-glucoside; CM, cow’s milk; CA, casein; WP, whey protein; AN, anthocyanin; Pt3G, petunidin-3-glucoside; Pn3G, peonidin-3-glucoside; BSA, bovine serum albumin; WPI, whey protein isolate; SPI, soy protein isolate; SPP, silkworm pupae protein; OVA, ovalbumin; SP, soy protein.

## 6. Effects of Protein–Anthocyanin Interactions on Protein Properties

Protein-bound anthocyanins not only affect the stability of anthocyanins but also affect the structural changes at all levels of proteins [93]. Many studies have found that anthocyanins easily interact with proteins in the actual production process, thus affecting the function and nutritional properties of proteins [108,130]. Proteins have excellent solubility, foaming stability, digestibility, and emulsifying properties. Anthocyanins can have a certain impact on the protein structure and functional properties during the processing of food, which can provide important references for improving the application of proteins.

### 6.1. Effects of the Interaction between Proteins and Anthocyanins on the Properties of Proteins

#### 6.1.1. The Effect of the Interaction between Anthocyanins and Proteins on the Solubility of Proteins 

The solubility of proteins is a prerequisite for their application in many foods and beverages. Solubility reflects the ability of proteins to interact with water molecules, and proteins with high hydration ability have better solubility compared with proteins with low hydration ability [131]. In addition, the solubility of proteins may in turn affect their functional properties including emulsification and foaming properties. A previous report found that the factors affecting protein solubility include internal factors (amino acid composition and amino acid sequence) and external factors (pH, temperature, and ionic strength) [132]. Additionally, non-covalent interactions, hydrophobic interactions, and salt bridges also affect protein solubility [133]. The combination of anthocyanins and proteins can promote the cross-linking of proteins, change the net charge of proteins, and affect their solubility. Fu et al. studied the interaction between C3G and ovalbumin (OVA) under different pH conditions by spectroscopic and molecular docking techniques. The results showed that the addition of C3G could decrease the surface hydrophobicity of OVA and improve OVA solubility [121]. Compared with soybean protein isolate (SPI), blueberry anthocyanins (BAs) could more significantly increase the solubility of the BAS-SPI complex. This may be due to the fact that BAs interacted with SPI to alter the protein structure (unwinding polypeptide chains and exposing hydrophobic groups) [134]. The interaction between black rice anthocyanins and CA was found to cause a red shift in the maximum absorption wavelength peak of tyrosine and tryptophan residues, which increased the micro-environment polarity of tyrosine and tryptophan residues and improved CA solubility [135]. Compared with purple potato protein (PPP), it was found that anthocyanins from purple potatoes better enhanced the UV absorption of PPP, which may be because the interaction between anthocyanins and protein can enhance the micro-environmental hydrophobicity of aromatic amino acids in PPP [136]. Zang et al. studied the interaction between WPI/BSA and BAS. The results showed that the solubility of WPI and BSA increased with the addition of BAS, which may be because the exposure of hydrophobic groups inside the WPI/BSA structure reduced the surface hydrophobicity and increased WPI/BSA solubility. In addition, the phenolic hydroxyl group in anthocyanins could also improve WPI/BSA solubility [119].

#### 6.1.2. The Effects of the Interaction between Anthocyanins and Proteins on the Foamability and Emulsification of Proteins

Foams are metastable systems dispersed in continuous phases at the interface of air and liquid, and foams are influenced by protein solubility, surface properties, and structural conformation [122]. Many studies have confirmed that anthocyanins can affect the foaming and emulsifying properties of the protein–anthocyanin complex. Compared with non-covalent binding, the covalent interaction between SPI and C3G better enhanced the emulsification and foamability of SPI, which was attributed to the alteration of the secondary structure of SPI by anthocyanins [137]. The complexation of SPI with anthocyanins changed the secondary structure of SPI and increased its solubility, thus improving its emulsifying and foaming properties [130]. The emulsification activity index (EAI) and emulsification stability index (ESI) represent the ability of proteins to form and stabilize emulsions at the oil/water interface, which is critical for the development of a variety of traditional and novel foods. Compared with natural rice protein (RP), anthocyanin complexation with RP was found to alter the conformation and interface properties of RP, thereby increasing EAI and ESI values of RP [116]. In another study, the addition of anthocyanins changed the conformation of WPI and improved the emulsification property of the WPI–anthocyanin complex [123]. Zang et al. studied the interaction between WPI/BSA and blueberry anthocyanins (BAs). The results showed that the non-covalent combination of WPI/BSA with BAS could improve the emulsification and solubility of WPI/BSA, which may be because the phenolic hydroxyl groups of BAS increased the affinity of WPI/BSA towards the oil–water interface [119]. The formation of the protein–anthocyanin complex increased the elasticity of the interface layer around oil droplets or bubbles, preventing bubble collapse or aggregation. Moreover, the protein–anthocyanin complex contributed to increasing the proteins solubility and promoting the rapid transfer of proteins to the interface layer, thus improving their emulsifying and foaming properties [138]. 

#### 6.1.3. The Effects of the Interaction between Anthocyanins and Proteins on Digestive Characteristics

Protein digestibility refers to the ratio of nitrogen absorbed by the human body from protein to nitrogen intake, which is an important index to evaluate the nutritional value of protein, reflecting the degree of protein decomposition and utilization by humans and animals [139]. The presence of anthocyanins can induce partial unfolding of protein structures and increase the accessibility of sensitive peptide bonds, thus improving their digestibility. The covalent binding of SPI with C3G was found to improve the gastrointestinal digestion characteristics of SPI and reduce the permeability of peptides in digested samples [138]. In another study, the addition of anthocyanins also increased the rate of digestion and absorption of SPI [124]. The gastrointestinal digestion model *in vitro* is an effective method to simulate the physiological conditions of the human gastrointestinal tract and investigate the structural changes and digestibility of food, and the results measured by the simulated digestion method *in vitro* have a good correlation with the experimental results *in vivo* [140]. Ma et al. studied the interaction mechanisms of SPI with mulberry anthocyanin (MA) and its effect on the digestibility of SPI through multispectral and gastrointestinal models *in vitro*. The results showed that MA could promote the digestion of SPI by gastric protease and slightly reduce the digestion rate of SPI in the intestinal fluid environment [141]. Many studies have indicated that anthocyanins in different natural resources can alter the secondary structure of proteins, increase the hydrolysis degree and soluble peptide components of energy spheres, and reduce particle size and aggregation, thereby improving proteins digestibility [125,126,142,143]. All in all, the interaction between proteins and anthocyanins can change the structure of proteins, increase the accessibility and permeability of sensitive peptide bonds, and improve the solubility of proteins in the intestinal fluid environment, thus enhancing their digestibility.

### 6.2. The Effect of the Interaction between Anthocyanins and Proteins on Protein Structure

The binding of anthocyanins to proteins can affect the structural changes in proteins. The amino acid residues of proteins contain tryptophan, tyrosine, and phenylalanine, which can produce intrinsic fluorescence at a certain excitation wavelength. Protein intrinsic fluorophore is highly sensitive to micro-environment polarity and has been widely used to monitor protein structural changes [138,144]. Wang et al. investigated the interaction between C3G and α-CA, β-CA, WP, and β-lactoglobulin by fluorescence spectroscopy, Fourier infrared spectroscopy, and circular dichroism. The results showed that C3G had a fluorescence static quenching effect on all four milk proteins, which was attributed to the fact that C3G could change the polarity of the surrounding environment of the tryptophan residues the milk proteins [129]. Purple potato anthocyanins were found to reduce the content of α-helix and β-corner in CA and WP, whereas they increased β-fold and irregular curl contents, thus effectively quenching the intrinsic fluorescence of CA and WP and implying that anthocyanins can effectively change the structure of CA and WP [100]. Meng et al. investigated the interaction between C3G and WPI by fluorescence spectroscopy. The results showed that C3G could induce static fluorescence quenching, which was attributed to the fact that C3G could change the secondary structure of WPI [103]. It was also found that the non-covalent binding of WPI and BAS could change the micro-environment of amino acid residues in WPI and its secondary structure [101]. Moreover, anthocyanins can be catalyzed by enzymes (polyphenol oxidase) or self-oxidized (such as in alkaline environments or the presence of oxidants) to produce electrophilic ortho-quinones, which undergo various continuous condensation reactions with other ortho-quinones, thereby changing the protein structure [132]. Sui et al. studied the non-covalent and covalent binding interactions between black rice anthocyanins and SPI and the effects of the interactions on the function and conformation of SPI. The results showed that the covalent binding of anthocyanins to SPI could decrease α-helix and β-fold contents and change the tertiary structure of SPI [130]. In another study, anthocyanins from red raspberry were found to change the secondary and tertiary structures of an SPI nanogel [127]. Currently, research on the conformational changes in proteins caused by anthocyanins mainly focuses on their impact on the secondary structure of proteins. However, there is relatively little research on the effects of anthocyanins on the tertiary and quaternary structures of proteins, which is attributed to the complex structure of proteins themselves. We should develop more methods to further clarify the impact of anthocyanins on the advanced structure of proteins in the future.

## 7. Application of the Interaction between Anthocyanins and Proteins

Anthocyanins show various biological activities including antioxidant, anti-cancer, anti-inflammatory, vision protection, weight loss, and other effects (Figure 6) [100,145]. Hence, the high-value application of anthocyanins has received widespread attention. Natural ingredients present in food interact with anthocyanins to improve anthocyanin stability and meet consumer demand for green additives. Anthocyanins are especially abundant in berries, vegetables, tubers, and grains. Protecting the stability of anthocyanins helps to enhance their application value. Many studies have confirmed that the interaction between anthocyanins and proteins can also protect anthocyanins to a certain extent [101].

### 7.1. Application of Anthocyanin–Protein Complexes in Liquid Systems

As people pay more and more attention to food nutrition, all kinds of new health drinks on the market continue to emerge endlessly. Additives including sucrose and vitamin C are often added in functional beverage systems, resulting in a decrease the anthocyanin stability. However, adding a certain amount of protein can improve the nutritional value of fruit juice and protect the stability of anthocyanins [128]. A previous report found that ascorbic acid could reduce the chemical and color stability of anthocyanins, whereas WPI could significantly increase the stability of anthocyanins in beverage systems. Moreover, WPI could increase the half-life of anthocyanins [110]. Wang et al. investigated the effect of preheated WPI on the color stability of a beverage model system in the presence of rose anthocyanin extract (RAE). It was found that preheated WPI at 100 °C could effectively improve the color stability of RAE in the beverage model system [126]. In another study, the total anthocyanins and physicochemical properties of the WP (4.0%) beverage formula did not show significant changes with storage time [146]. WP drinks containing anthocyanins from *Robinia pseudoacacia* peel have good stability and great market potential. The addition of heat-denatured WP was found to significantly enhance the stability of anthocyanins during storage. This may be attributed to the formation of the complex between anthocyanins and WP through hydrogen bonding, enhancing the stability of anthocyanins in the presence of ascorbic acid [147]. At present, milk powder and anthocyanins extract are used as the main raw materials. A characteristic yogurt product with both nutrition and flavor was developed using *Lactobacillus bulgaricus* and *Streptococcus thermophilus* as starter cultures [148]. Quan et al. investigated the effects of β-cyclodextrin (β-CD), WP, and SP on the color loss and degradation of anthocyanin from purple sweet potatoes (PSPA) in a model beverage system during heat treatment and storage. The results showed that SP (25 mg/L) could improve the color and thermal stability of anthocyanins when heated at 100 °C for 30 min. In addition, the addition of β-CD (2500 mg/L) could significantly enhance the storage stability of PSPA [116]. In another study, the complex formed by bovine serum protein and BAS could effectively increase the stability of anthocyanins and improve the functional properties of proteins in a fruit milk system [116]. These results provide a theoretical basis for the processing of fruit milk products.

### 7.2. Application of Anthocyanin–Protein Complexes in Solid-State Systems

Anthocyanins contribute to improving the sensory indicators of protein-containing foods and increasing the nutritional function of foods. Compared with other plant proteins, rice protein is a nutritive protein with lower allergenicity and higher amino acid composition, which is suitable for most people [149]. In addition, the non-covalent binding (hydrogen bonding and hydrophobic interactions) of rice proteins to anthocyanins can improve the antioxidant activity and foamability of rice proteins [122]. Therefore, rice protein has a very good application prospect in products such as infant formula and baked goods. Milinic et al. investigated the effect of anthocyanins from grape pomace on the antioxidant properties of goat milk powder. The results showed that the antioxidant properties of goat milk powder could be significantly improved by adding anthocyanins during heat treatment [150]. Previous research found that the combination of anthocyanins in bayberry juice and pea protein could decrease the α-helix content by 1.22%, the β-folding content by 1.98%, and the random curling content by 0.52% in pea protein, whereas it increased the β-angle content by 3.72% and the thermal stability of pea protein, implying that anthocyanins can change the structure of pea protein [151]. Wu et al. prepared a BAS-SPI complex and treated it by high-power pulse microwave (HPM) to obtain an HPM-BAS-SPI complex, which was then applied to cake. It was found that the HPM-BAS-SPI complex could effectively inhibit water loss during cake baking and improve the hardness and chewability of the cake, which played an important role in improving the shelf life and quality of the cake [134]. In summary, protein–anthocyanin interactions can affect temporal functional properties, which provides an important reference for designing multi-functional and high-quality foods.

## 8. Conclusions and Prospects

Anthocyanins are widely used in food, medicine, cosmetics, and other fields because of their wide source, good physiological activities, and rich color. With the continuous development of science, the extraction and purification methods for anthocyanins are becoming more and more mature. The main factors affecting the extraction of anthocyanins are the characteristics of the sample substrate and extraction parameters (pH value, solvent, temperature, time, etc.). At present, the extraction method has developed from the traditional solvent method to deep eutectic solvent extraction, ultrasound-assisted extraction, microwave-assisted extraction, supercritical fluid extraction, and other combined technologies. Compared with traditional solvent extraction, the potential extraction methods exhibit some advantages including higher extraction rates, lower energy consumption, and shorter extraction times. Nevertheless, the potential extraction techniques also show some disadvantages including high extraction cost and parameters that are difficult to control. From the perspective of industrialization, microwave- and ultrasound-assisted extraction methods have been applied in industry, while supercritical fluid extraction and high-voltage pulse electric field-assisted extraction methods have difficultly in achieving the large-scale industrial extraction of anthocyanins because of high equipment costs and strict extraction conditions. We should develop more combined extraction methods for large-scale, rapid, and efficient extraction of anthocyanins from natural resources in the future.

Anthocyanins are good natural coloring agents that exhibit various biological activities including antioxidant, anti-inflammatory, anti-cancer, and hypoglycemic activities. The poor stability and easy degradation of anthocyanins limit their application in food, cosmetics, medicine, and other fields. In recent years, the interaction between anthocyanins and proteins has become a focus of research. Many studies have indicated that the interaction between anthocyanins and proteins could enhance the stability of anthocyanins, improve the foaming and emulsifying properties and digestibility of proteins, and endow them with higher nutritional value. Currently, the complex formed by proteins and anthocyanins from different sources is being widely used to improve the sensory properties of food and develop new functional foods. Numerous studies have confirmed that the interactions between some polyphenols and proteins have a certain negative impact on the bioavailability of polyphenols and the digestibility of proteins. However, there few studies have drawn similar conclusions regarding the interaction between anthocyanins and proteins. Thus, we need to conduct more comprehensive research on the impact of the interactions between anthocyanins and proteins on their structural and functional properties in the future. This research can provide a basis for improving the processing conditions and parameters of food and provide important references for developing new types of food with more nutritional value. With the development of technology, the techniques used to characterize the interaction mechanisms between large and small molecules are constantly being updated. However, the current research on the interactions between anthocyanins and proteins using these new technologies is still very limited. A single technology cannot fully analyze the structural changes in proteins. We urgently need to combine multiple analytical methods to study the interaction potential mechanisms between proteins and anthocyanins deeply and further expand their application fields in the future.

## Figures and Tables

**Figure 1 molecules-29-02815-f001:**
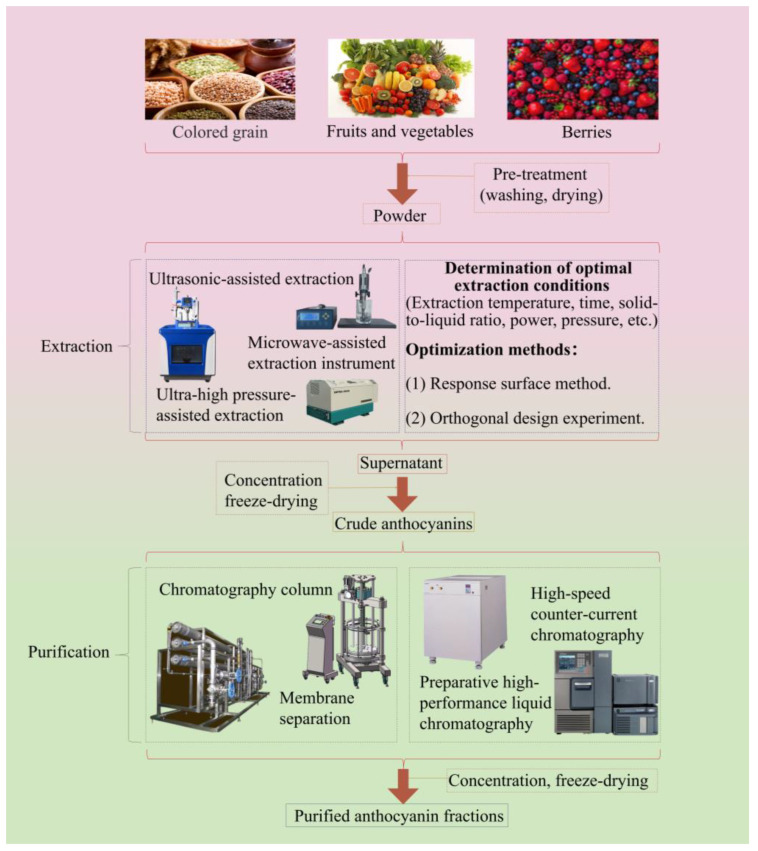
Different methods for the extraction and purification of anthocyanins from natural resources. Abbreviations: ultrasound-assisted extraction (UAE); microwave-assisted extraction (MAE); ultra-high pressure-assisted extraction (UHPAE); response surface methodology (RSM); column chromatography (CC); membrane separation (MS); high-speed counter-current chromatography (HSCCC); preparative high-performance liquid chromatography (PHPLC).

**Figure 2 molecules-29-02815-f002:**
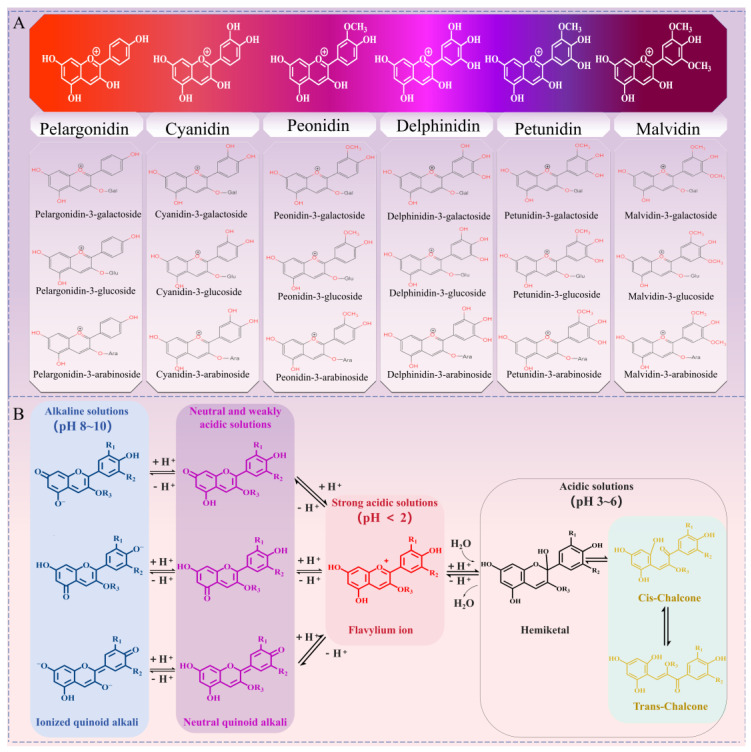
Common structure of anthocyanin monomers and color changes in anthocyanins at different pH values. Note: (**A**) Anthocyanin monomers and their structures; (**B**) Anthocyanin monomers at different pH values.

**Figure 3 molecules-29-02815-f003:**
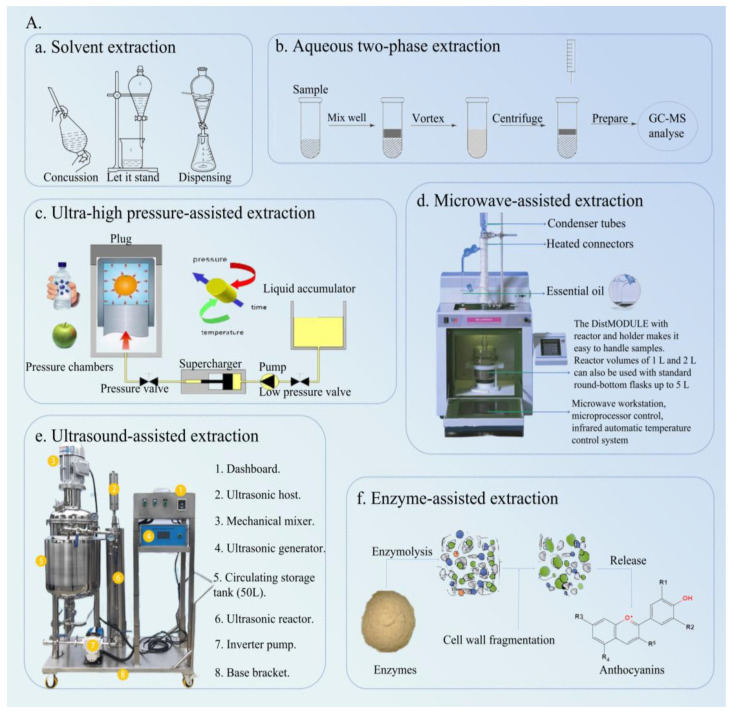
The principle of different extraction (**A**) and purification methods (**B**) for anthocyanins.

**Figure 4 molecules-29-02815-f004:**
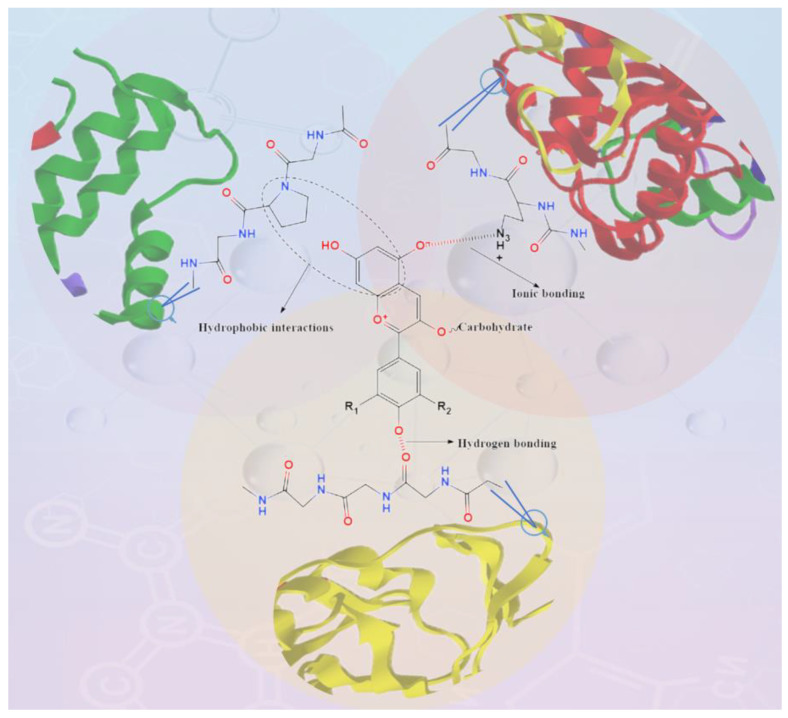
Non-covalent binding mechanisms of proteins and anthocyanins through hydrogen bonding, hydrophobic interactions, and ionic interactions.

**Figure 5 molecules-29-02815-f005:**
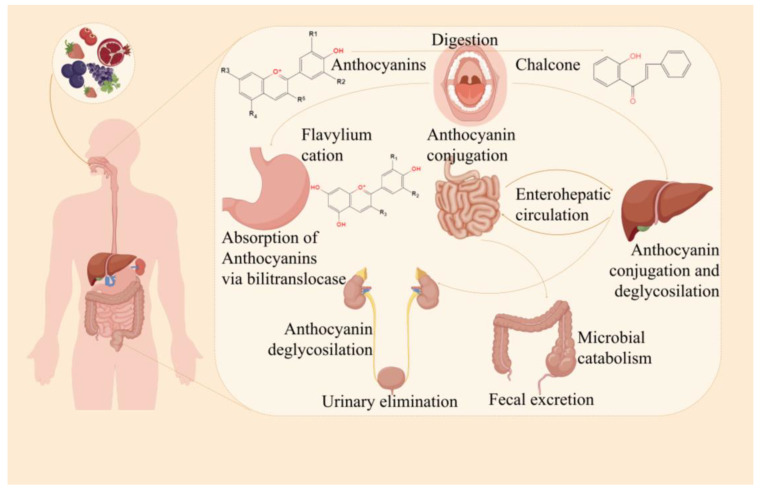
The digestive process of anthocyanins in the human body.

**Figure 6 molecules-29-02815-f006:**
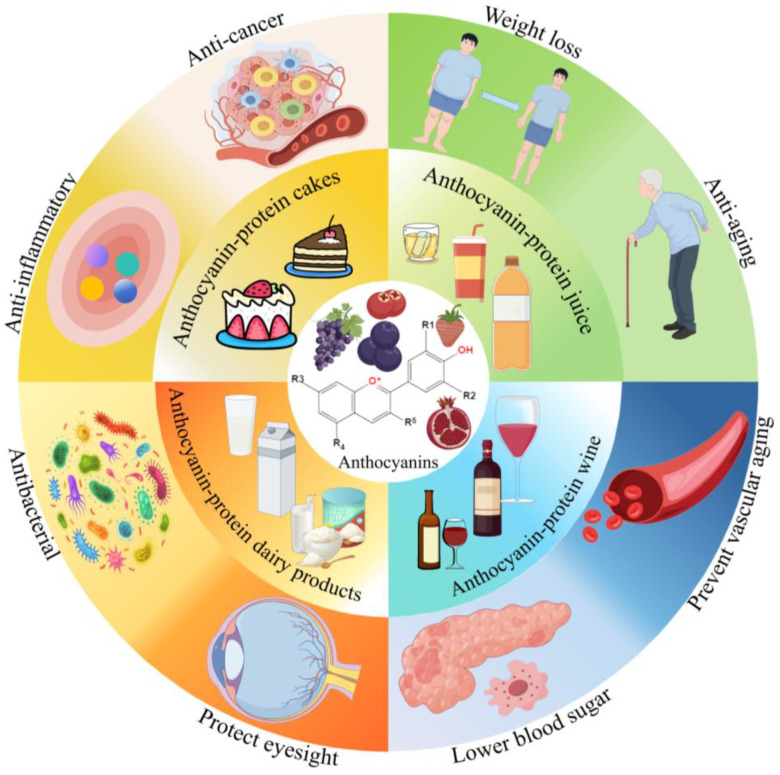
Application and function of protein–anthocyanin functional products.

## Data Availability

All data generated or analyzed during this study are included in the article.

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
