# Peer review of "Research Progress on the Extraction and Purification of Anthocyanins and Their Interactions with Proteins"

_molecules, 2024, doi:10.3390/molecules29122815_

Round 1

Reviewer 1 Report (Previous Reviewer 2)

Comments and Suggestions for Authors

The manuscript was corrected appropiatelly 

Author Response

  1. Response to comment:(The manuscript was corrected appropiatelly.)

Response: We sincerely appreciate the valuable comments provided by the reviewer on the manuscript, and at the same time, we sincerely appreciate your recognition of the manuscript.

Special thanks to you for your good comments.

Reviewer 2 Report (Previous Reviewer 3)

Comments and Suggestions for Authors

I have already recently reviewed the original version of the manuscript entitled "Research progress on the extraction and purification of anthocyanins and their interaction with proteins".

Now I have  reviewed the improved version and I found that, the authors made some changes in the manuscript in order to improve it without any changes on the content and framework of the paper. therefore, the manuscript has been significantly improved and now warrants publication in molecules, and no further revisions are required.

Author Response

  1. Response to comment:(I have already recently reviewed the original version of the manuscript entitled "Research progress on the extraction and purification of anthocyanins and their interaction with proteins".Now I have  reviewed the improved version and I found that, the authors made some changes in the manuscript in order to improve it without any changes on the content and framework of the paper. therefore, the manuscript has been significantly improved and now warrants publication in molecules, and no further revisions are required.)

Response: On behalf of all the authors, I express my sincerest gratitude to you. Thank you for your constructive opinions on the manuscript, which is beneficial for significantly improving its quality.

Special thanks to you for your good comments.

Reviewer 3 Report (Previous Reviewer 4)

Comments and Suggestions for Authors

Authors have nicely implemented the changes requested.

However, still important issues remain.

Figure 2A: SWAP the column with Peonidin and Delphinidin. The color associated to Delphinidin is lillac and with peonidin is pale purple.

Figure 2B and lanes 117-118 "Anthocyanins exist in the form of colorless methanol off-base or hemiacetals within the pH range of 3-6" 

This is true for the aglycons, which does not exist in nature. You always find glycosylated or glyco-acylated anthocyanins that in the range of pH 3-6 do have a color that change at higher pH.

Please try to change this sentence. 

You instead referred to other bibliography which is not adeguate to your sentence in lanes 117-118.

Author Response

  1. Response to comment:(Authors have nicely implemented the changes requested. However, still important issues remain. Figure 2A: SWAP the column with Peonidin and Delphinidin. The color associated to Delphinidin is lillac and with peonidin is pale purple.)

Response: We first thank you for your valuable comments. Based on your opinion, we have made revisions to the issue you mentioned. In the revised manuscript, the revised sections is marked in red to easy check.

  1. 2. Response to comment:(Figure 2B and lanes 117-118 "Anthocyanins exist in the form of colorless methanol off-base or hemiacetals within the pH range of 3-6".This is true for the aglycons, which does not exist in nature. You always find glycosylated or glyco-acylated anthocyanins that in the range of pH 3-6 do have a color that change at higher pH. Please try to change this sentence. You instead referred to other bibliography which is not adeguate to your sentence in lanes 117-118.)

Response: Thanks for the Reviewer’s suggestion. According to the Reviewer’s suggestion, we have revised the relevant contents as follows: Anthocyanins exist in the form of colorless methanol off-base or hemiacetals within the pH range of 3-6 (Fig. 2B). Notably, most glycosylated or glycosylated anthocyanins exhibited high color stability in the pH range of 3-6, while the color stability of anthocyanins decreased at high pH values. In addition, we also add relevant supporting literature. In the revised manuscript, the revised sections is marked in red to easy check.

Special thanks to you for your good comments.

Reviewer 4 Report (Previous Reviewer 5)

Comments and Suggestions for Authors

The points raised by this reviewer have been addressed

Author Response

  1. Response to comment:(The points raised by this reviewer have been addressed)

Response: We sincerely appreciate your recognition of the manuscript. We will continue to make efforts to conduct relevant research.

Special thanks to you for your good comments.

This manuscript is a resubmission of an earlier submission. The following is a list of the peer review reports and author responses from that submission.

Round 1

Reviewer 1 Report

Comments and Suggestions for Authors

The authors clearly made significant efforts to write the review. The paper is well structured and covers a wide array of information related to anthocyanin extraction and purification. The schematic descriptions of the methods is well done given that they facilitate the visualisation of the method in parallel to the available information in the text. The scope of the paper fits well the scientific scope of the journal, and English level is good. I recommend that the paper be published in the journal

Reviewer 2 Report

Comments and Suggestions for Authors

Is an interesting and complete review

Reviewer 3 Report

Comments and Suggestions for Authors

Dear Authors

The authors evaluated the manuscript entitled ‘Research Progress on the Extraction and Purification of Anthocyanins and Their Interaction with Proteins‘. Overall the manuscript is well-written and presents an interesting topic on the extraction and purification of anthocyanins, and their interactions with proteins. But some corrections must be done before possible consideration for publication in the Journal as detailed in the following:

ý  Page 1, Lines 41-43: However, this method exhibits some disadvantage including low extraction efficiency, long time consumption, and high solvent consumption....... Should be....... However, these methods exhibit some disadvantage including low extraction efficiency, long time consumption, and high solvent consumption.

ý  Page 2, Line 52: Hence,   the crude.....Delete the space

ý  Page 3: Figure 1. Determination ofoptimal extraction  ...... Should be... Determination of optimal extraction

ý  Page 3: Figure 1. Solid – to – liguidratio  ...... Should be... Solid – to – liguid ratio

ý  Page 3: Figure 1. There are some abbreviations such as UAE , MAE, ...etc in figure 1 .....Should mentioned in the title of the figure what these abbreviations means?

ý  Page 3, Line 63: Figure 1. Different methods for extraction and purification anthocyanins from natural resources. Should be Different methods for extraction and purification of anthocyanins from natural resources.

ý  Page 3, Line 65: factors (such as temperature, pH value, oxygen, light, ascorbic acid, and metal ions), Remove the brackets

ý  Page 7, Line 144: (such as ......)  Remove the brackets

ý  Page 7, Line 157: has the some advantages..... Delete "the"

ý  Page 7, Line 159 and line 161: Therefore, SE has certain limitations..... SEM not SE.... to be uniform in whole text

ý  Page 11, Line 189 : Yigit et al. Extracted ......the reference needs number (check journal requirements).

ý  Page 12, Lines 225, 226: The IC50 values of anthocyanin extracts for scavenging DPPH and OH radicals were 0.026 mg/mL and 0.406 mg/mL, respectively.... I would suggest authors to add this sentence to be more explained as follow :

 Moreover, their antioxidant potential are influenced by numerous parameters intrinsic and extrinsic such as light exposure, sample pretreatment, temperature, storage, and oxygen (Mohdaly et al., 2022). Therefore the following paper should be added as a reference

Mohdaly, A.A.A.; Roby, M.H.H.; Sultan, S.A.R.; Groß, E.; Smetanska, I. Potential of Low Cost Agro-Industrial Wastes as a Natural Antioxidant on Carcinogenic Acrylamide Formation in Potato Fried Chips. Molecules 2022, 27, 7516. https://doi.org/10.3390/

ý  Page 24: Figure 6. Protect eyesight (repeated twice) please delete one of them

ý  Applications of anthocyanin-protein interactions are only briefly mentioned and not explored in depth.

Reviewer 4 Report

Comments and Suggestions for Authors

The review manuscript is too long and tend to mislead the reader to what is really important message of the manuscript.

Many scientific imperfections are present and almost no test to discuss the figures.

The review manuscript is too long and the focus is readly lost in the differect sections.

Many scientific imperfections:

1) lanes 97-98: "Pelargonidin, delphinidin, petunidin, cyanidin, peonidin, and malvidin are commonly used as monomers of anthocyanins." Wrong they never are found as monomeric aglycons but always glycosydated and acylated.

2) lanes 111-112 "Anthocyanins exist in the form of colorless methanol off base or hemiacetals within the pH range of 3-6." That is not true in that range of pH the color can be still kept and pH 4.5 is the industry golden standard to measure their absorbance.

3) In Figure 1: no descriptions for the abbreviations. The reader has to read much below to get those meaning. Still almost no text for figure1.

4) In figure 2A: there is a switch between peonidin and delphinidin. The figures are swapped, as well as peonidin should come before delphinidn as color and glycosides. No text for figure 2 after this complex description of their chemical structure and the panel B, almost no info in the text.

4) Figure 3: still not enough text.

table 1 : too long !

5) The description of the extraction and purification methods are also very long.

6) Figure 5 no esplanation on what you see in the figure!

7) Figure 7: idem!

Overall, the manuscript can be splitted into two: one on the extraction and purification methods and the second on the interaction of the anthocyanins with the proteins (liquid and solid phase).

My suggestion is to reject the paper in the present form and split, resubmitt the two sugested paper out of it.

Reviewer 5 Report

Comments and Suggestions for Authors

The topic of the manuscript is interesting, especially the section focused on protein-anthocyanin interactions.

My suggestions are as follows.

In the graphical abstract there are different typos such as: supernatant, liquid ratio, of optimal. Please check.

Line 69

Change “microcapsules” to “encapsulation strategies”

Line 97

Change “Pelargonidin” to “pelargonidin”

Table 1

It would be of interest to know the extraction yield as percentage with respect to a standard protocol with solvent.

Moreover, I suggest to report the Latin names of the compounds. Finally, I suggest adding a footnote with indicated the abbreviation used, in order to make the table self-standing

The section dedicated to protein-anthocyanin interactions is the most attractive. In the other hand, I suggest to summarize the information with a Table, in order to make it more direct and favour comparison.

 The binding energy between protein and anthocyanin or the dissociation constant should also be reported.

Comments on the Quality of English Language

I suggest revision pf Emglish language